# Encouraging Sustainable Consumption through Gamification in a Branded App: A Study on Consumers' Behavioral Perspective

Chih-Wei Lin [1], Chun-Yu Chien [1], Chi-Pei Ou Yang [2] and Tso-Yen Mao [1,*]

[1] Department of Leisure Services Management, Chaoyang University of Technology, Taichung City 413, Taiwan
[2] Department of Business Administration, Chaoyang University of Technology, Taichung City 413, Taiwan
* Correspondence: tymao.research@gmail.com; Tel.: +886-4-2332-3000 (ext. 7468)

**Abstract:** Gamification, an innovative tool for interacting with consumers that can be seen as a new trend in marketing, could enhance customer behavior, such as greater loyalty. This paper investigates the relationship between gamification, attitude, and customer behavioral intention. Using the Starbucks branded app as the tool, this study aims to identify the influencing factors of the app from a gamification perspective (achievement, challenge, rewards) on customer engagement, perceived playfulness, attitude, and behavior. The Technology Acceptance Model and Mechanics Dynamics Emotions were employed, using perceived playfulness as an intrinsic motivation of the Technology Acceptance Model. The subjects of this study were 581 customers in Taiwan who have used the Starbucks branded app. The primary data were gathered to test the hypothesis and propose a model. The findings showed that game elements have a positive influence on customer engagement and perceived playfulness. Gamification positively enhances customer engagement on the Starbucks branded app and creates joyful emotion and sustainable consumption. Thus, the game element positively affects subsequent behaviors, such as attitude and behavioral intention.

**Keywords:** game element; customer engagement; perceived playfulness; attitude; behavioral intention





## 1. Introduction

With the popularity of smartphones, users are paying more attention to experience and perception. Meanwhile, branded apps have been recognized as an effective way to build engagement between the brand and the customers [1]. Besides promoting the brands, the branded apps can also strengthen brand interaction and build brand image and awareness. Moreover, companies are seeking and creating communication platforms to motivate employees and increase consumer satisfaction and loyalty [2]. Goldman (2010) mentioned that many companies use branded apps to replace traditional membership cards [3]. This practice follows Sustainable Development Goal (SDG) 12, which ensures sustainable consumption and production patterns—as in Target 12.5, "substantially reduce waste generation through prevention, reduction" and Target 12.6, " . . . adopt a sustainable practice and to integrate sustainability information into their reporting cycle"—because apps can encourage consumer retention related to value-added service [4]. Furthermore, branded apps are an effective tool for customers to received update information on companies' promotions, contents, and services at anytime and anywhere [5].

Comprehensive innovation training is essential to keep up with or even outperform competitors [6]. According to Chen, Liu, and Dai (2013), branded apps could help companies with brand building [7]. Branded apps create a value that differs from paid apps as they aim to facilitate customer engagement with brands, unlike paid apps that aim to earn profit and revenue by providing services [8]. Hence, many companies, especially chain store brands such as McDonald's, Pizza Hut, and Starbucks, have successively launched their own branded apps. Companies have also begun connecting their membership systems to the branded apps. Among those apps, Starbucks' branded app, in which Starbucks

guests account for nearly two-thirds (61.4%) of the users [9,10], is the best developed app with the most rewards program among a list of major restaurant chains [11].

Gamification has become a popular element incorporated into individuals' daily activities [12] and used in education, utilities, transportation, etc. [2]. It can be seen as a new trend to enhance user engagement in various contexts [13], such as e-learning, marketing, eco-sustainability, medical health, and tourism [14]. Gamification is also a useful tool to encourage user immersion and adapt to related content, and an innovative and sustainable alternative to maintaining social benefit [15]. According to Treiblmaier et al. (2018), gamification can be defined as the use of game-design elements in any non-game system context to increase users' intrinsic and extrinsic motivation [16]. The game design elements can enhance consumers' knowledge, attitudes, and behavioral intention [17]. The Starbucks' branded app allows customers to accumulate stars and receive the latest news. It has the characteristic of game elements that create fun and valuable information, and help placement from Starbucks. Gamification can foster individuals' motivation and change their behaviors [18,19]. Therefore, this paper hypothesizes that gamification is one of the primary keys that motivate customers to continue using the Starbucks' branded app.

The Technology Acceptance Model (TAM) is a representative model in information technology. It can effectively predict the behavioral probability of individuals adopting new technological systems. Turner et al. (2010) suggested that TAM could effectively predict an individual's behavior when a particular technology is first used [20]. TAM includes five variables: the perceived ease of use, perceived usefulness, attitude toward use, behavioral intention to use, and actual use. The two most significant factors in the model are the perceived ease of use and perceived usefulness, which are the core variables of TAM. However, the Starbucks' branded app includes game elements; the original TAM can no longer meet this research condition. According to TAM, when users encounter a new technology, perceived usefulness and the perceived ease of use are the two main factors. However, because the Starbucks branded app is very mature in Taiwan, its perceived usefulness and the perceived ease of use have been incorporated into the basic elements of software design. Therefore, more emphasis is placed on customer experience in terms of marketing strategies.

According to the gamification-related literature, Mechanics Dynamics Aesthetics (MDA) is the most representative framework. Li et al. (2019) suggested that Mechanics Dynamics Emotions (MDE) could explore user experience in gamification studies [21]. Mullins and Sabherwal (2020) noted that MDE is the only gamification framework that explicitly considers emotion as a critical factor [22]. Therefore, this study took MDE as the research framework. The MDE framework was adapted from the MDA framework [23]. Aesthetics is replaced with emotion in game design because aesthetics often arouse players' emotional responses, while interacting with the game. On the other hand, emotions can express customers' participation in behavior and results more effectively.

According to Schell (2008), mechanics is the most critical factor in gamification, and game elements are the basis that guides the entire game process [24]. Therefore, this study used game elements to represent the mechanism herein. Gamification dynamics are player behaviors that occur when users participate in games [25]; in other words, the behavioral responses generated when users interact with the game, prompting players to join or leave. Since customer engagement can be used to represent a specific interaction between a customer and a brand [26], this study used customer engagement to represent dynamics. Agarwal and Venkatesh (2002) pointed out that from the perspective of human-computer interaction, the influence of emotions on users mainly comes from the users' assessment and manipulation of tasks and media [27]. Moon and Kim (2001) argued that when individuals are trying out a new system, they would be willing to spend more time using it if they have a higher interest in the system [28]. In this case, the individual's positive perception of system efficiency could be enhanced; hence, this study used perceived playfulness to represent emotion.

A branded app is a marketing tool that retains engagement between brands and customers. To provide enjoyable customer experiences through apps, companies should understand the factors that prompt or hinder consumers from using apps. This study suggested that the Starbucks branded app has been using game elements, such as challenge, achievement, feedback, and emotion, in non-game situations to achieve its marketing concept.

When a software technology is close to maturity, it crosses the gap proposed by the innovation diffusion theory, and reaches a generalized level. The Starbucks branded app has been launched for over five years; at this stage, the marketing strategy should pay more attention to the user experience. As various game elements have different effectiveness in motivating consumers to engage in services [29], this study suggested that game elements are an important factor to the user behavior in using the Starbucks branded app.

Therefore, the objective of this study is to investigate the influencing factor of the Starbucks branded app from a gamification perspective (achievement, challenge, rewards) on customer engagement, perceived playfulness, attitude, and behavior. This study combined emotion (perceived playfulness) and dynamics (customer engagement) into TAM to fill the research gap on the interaction between the system and the customers. This study used the MDE framework to explain the effects of game elements on customer perceptions and behavior intention. Based on the results, managerial implications are provided for brand companies to implement marketing strategies, and enhance customer interaction and direction for future gamification research.

## 2. Literature Review and Hypotheses Development

Seaborn and Fels (2015) defined gamification as the intentional use of game elements for a gaming experience of non-game tasks and contexts [30]. Hofacker et al. (2016) mentioned that gamification could encourage individuals to create value behavior, such as greater loyalty [31]. Hamari and Koivisto (2015) pointed out that gamification is not like playing games, but is designed for entertainment and leisure. Gamification typically includes point systems, levels, or badges, and individuals participating in a specific task [32]. The main characteristic of gamification is "to compete, challenge and socially interact", which would often change customer behavior [24]. Several studies proposed the principle of gamification and the game design approach, such as the MDA framework and MDE framework.

### 2.1. MDE Theory

The MDA framework was proposed by Hunicke, LeBlanc, and Zubek (2004), focusing on mechanics, dynamics, and aesthetics [23], as well as exploring game design from a systematic perspective [33]. Robson et al. (2015) suggested that emotion is a better option for the results; therefore, the MDE framework is more commonly used for MDA within gamification [25]. MDE explores how gamification affects the user experience [21]. In particular, the MDE theory incorporates mechanics, dynamics, and emotions as interdependent aspects to better distinguish the emotional aspects of engagement outcomes related to gamified experiences [25]. Robson et al. (2015) claimed that gamification elements include mechanics, dynamics, and emotions [25].

### 2.2. Mechanics

Schell (2008) indicated that mechanics is the main element in games [34]. It can constrain players' behavior through the procedures and rules of the game [35]. De Mira Gobbo et al. (2021) proposed that the game's mechanics includes the game's rules, levels, and structure [36]. Hamzah et al. (2015) suggested that game mechanics have features, such as points, levels, leaderboards, virtual goods, badges, gifts, and charity [37]. According to Hunicke et al. (2004), game mechanics include achievements, collections, and badges, among others. Since mechanics is the main element in the game, this study focused on gamification mechanics [23].

The winning key of games is reward; without reward, players would lose their motivation to play games [38]. Flatla et al. (2011) pointed out that a challenge is related to reward [39]. When players overcome the challenge, they receive a reward. Then, feedback allows users to know their progress and performance in a gamified environment [39]. Feedback is like progress tracking, or a way to determine the objective of winning conditions [38]. Groening and Binnewies (2019) considered achievements a secondary reward system, which prompt the users to a specific behavior to obtain the achievement as a reward [40]. Previous studies suggested that achievement is a high priority that is related to game elements [30,32,41].

Therefore, this study considered achievements, challenges, and feedback as the feature of the gamification elements in the Starbucks branded app.

### 2.3. Dynamics

According to Robson et al. (2015), dynamic refers to the behavior that occurs when a player participates in the activity, which is how the player interacts with the implemented game mechanism [25]. Hamzah et al. (2015) proposed that game dynamics include rewards, status, achievements, competition, and self-expression [37]. Werbach and Hunter (2012) suggested that challenges, time pressure, personal expression, and dramatic tension represent game dynamics [41]. Mullins and Sabherwal (2020) pointed out that dynamics are challenging to predict, but it is through dynamics that the unintended consequences of gamification appear [22]. Therefore, the concept of game-challenge reactions was adopted in this study to identify game dynamics and defined behaviors toward overcoming difficulties while using the Starbucks branded app.

Further, when using the Starbucks branded app, mechanisms (e.g., tracking the collected points or star-level challenges), discounts, and gift redemption are used to stimulate user engagement. Darejeh and Salim (2016) proposed that companies incorporate gamification mechanisms (e.g., challenges, tasks, achievements and rewards, and winning conditions) into game design to fully understand the game content, implement customer experience, and promote customer engagement [42]. Harwood and Garry (2015) found that customer engagement would change if the system adds in-game elements [43]. Considering that customer engagement can be used to represent a specific interaction between a customer and a brand [26], this study used customer engagement to describe dynamics.

### 2.4. Emotions

According to MDE, the gamification mechanism affects users' emotions [25]. Game emotions are the psychological and emotional states triggered by players participating in the games. The purpose of mechanism and dynamics in gamification is to stimulate emotions [23]. Gamification can influence behaviors through rewards, thereby producing happiness and satisfaction. Games are inherently motivating because they are viewed as activities that people create for joy [12]. Hunicke et al. (2004) pointed out that games can stimulate emotional responses that produce enjoyment, which is an important goal for users to play the games [23]. Stepanovic and Mettler (2018) found that users' attention to the game system is satisfaction and playfulness [44]. Since the Starbucks branded app is hypothesized to produce a sense of playfulness due to rewards, achievements, and activity challenges, this study used playfulness to represent emotions and defined playfulness as inner emotional reactions and expressing feelings while using the Starbucks branded app.

### 2.5. Perceived Playfulness and Attitude

According to Venkatesh (2000), perceived playfulness refers to an individual's perception of a specific system, whether it is enjoyable aside from any performance resulting from system use [45]. Additionally, perceived playfulness is known as intrinsic motivation [46], shaped by the individual's experiences with the environment [28]. While users are in a state of playfulness, they find the interaction intrinsically interesting, and perceive pleasure and

enjoyment rather than extrinsic rewards [28]. Prior studies found that perceived playfulness positively affects attitude and behavior intention [47–50].

Pleyers and Poncin (2020) suggested that attitude toward service providers is positively affected by playfulness [51]. Priyadarshini et al. (2017) found that perceived playfulness evokes the website attitude of individuals [49]. Hwang and Choi (2020) found that in a gamification loyalty program, playfulness has a positive impact on attitude [52].

### 2.6. Customer Engagement, Attitude, and Behavior Intention

Hollebeek et al. (2019) defined customer engagement as a motivational investment of cognitive, emotional, behavioral, and social knowledge/skills in brand interactions [53]. Hollebeek et al. argued that customer engagement is the resource that customers invest in brand interactions. Customer engagement stimulates individuals to invest more of their personal resources in perceived value-adding interactions [54]. Prior studies also found that customer engagement is a state of mind that drives customer behavior [55,56]. Customer engagement has emerged as an influential concept in marketing and a motivator that drives behavior [57].

Bergel et al. (2019) found that customer engagement can positively affect attitude, leading to increased positive customer behavior in the future [58]. Customer engagement has a positive increment in behavioral responses, such as customer loyalty, word-of-mouth (WOM), and customer feedback. Yen et al. (2020) proposed that customer engagement positively relates to customer participation behaviors [59]. Molinillo et al. (2020) claimed that customer engagement positively affects their repurchase intention in social commerce websites [60].

### 2.7. Attitude and Behavior Intention

The Theory of Reasoned Action (TRA) (Fishbein and Ajzen, 1977), the Theory of Planned Behavior (TPB) (Ajzen, 1991), and the TAM (Davis et al., 1989) are well-known theories suggesting that attitude has a positive influence on behavior. According to Fishbein and Ajzen (1977), attitude is a learned predisposition of humans that leads to a reaction toward objects, ideas, or opinions. The influence of attitudes on customer behavior has become a concern for researchers in customer behavior studies [61]. Ketelaar and Van Balen (2018) pointed out that the influence of attitude on behavior depends on how attitude is formed [62]. Glasman and Albarracín (2006) argued that attitude directly affects an individual's behavior. Prior studies confirmed that attitude has a positive influence on behavioral intentions [62,63]. For instance, Kim et al. (2021) found that attitude positively influences an individual's behavior [64]. Han et al. (2019) maintained that a higher degree of attitude forms a greater degree of behavioral intention [65]. Hamari and Koivisto (2015) also posited that favorable attitudes toward gamification lead to continued use of a gamification service [32].

According to the previous literature review, this paper proposes the conceptual framework and hypotheses as follows (Figure 1):

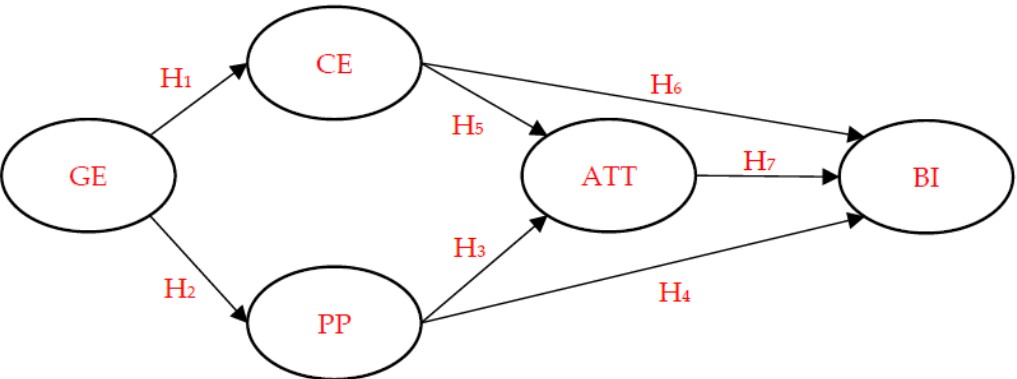

**Figure 1.** Conceptual framework and hypotheses.

**H$_1$.** *Game elements have a positive effect on customer engagement.*

**H$_2$.** *Game elements have a positive effect on perceived playfulness.*

**H$_3$.** *Perceived playfulness has a positive effect on attitude.*

**H$_4$.** *Perceived playfulness has a positive effect on behavior intention.*

**H$_5$.** *Customer engagement has a positive effect on attitude.*

**H$_6$.** *Customer engagement has a positive effect on behavior intention.*

**H$_7$.** *Attitude has a positive effect on behavior intention.*

### 3. Materials and Methods

*3.1. Research Model Proposal*

To measure the influences of gamification on consumption behavior, this study selected several variables: MDE from Robson et al. (2015) [25]; customer engagement from Van Doorn et al. (2010) [66]; perceived playfulness from Moon and Kim (2001) [28]; attitude from Dick and Basu (1994) [67]; behavior intention from Fishbein and Ajzen (1977) [61].

*3.2. Sampling Design and Data Collection*

According to McLeod (2019), quantitative research design is suitable for numerical measurement [68]. This study employed the quantitative method using Google forms to collect primary data from the target audience. The survey was divided into two parts: the demographic characteristics of the participants and 35 scale indicators associated with the researched constructs. A seven-point Likert scale, ranging from 1 (strongly disagree) to 7 (strongly agree) was used to measure the research model indicators. The scale of game elements was adapted from Hamari (2015) [32], and Eckardt and Robra-Bissantz (2018) [69]; the scale of customer engagement was modified from Vivek, Beatty, and Morgan (2012) [70]; the scale of perceived playfulness was based on Moon and Kim (2001) [28]; the scale of attitude was adapted from Dick and Basu (1994) [67]; the scale of behavior intention was based on Fishbein and Azjen (1969) [71].

A total of 650 questionnaires were distributed to the users of the Starbucks branded app via the Internet from 25 January 2021 to 15 February 2021 by snowball sampling. After eliminating the invalid samples, there were 581 valid samples, with an effective return rate of 89.38%. The response rate was acceptable based on the 80% rate suggested by Fryrear (2015) [72].

*3.3. Analysis Method*

To examine the relationship and the causal effects of the proposed model, the data were analyzed using structural equation modeling (SEM) and AMOS. According to Awang (2015), SEM is the second generation of multivariate analysis and the favored method among researchers in analyzing data [73]. CFA is the basis of the measurement model in full SEM, and can be estimated using SEM software. AMOS is a popular software that can produce almost the same statistics as Mplus and LISREL, and has the feature of reading an SPSS data file directly [74]. Therefore, using SPSS AMOS to obtain SEM results is appropriate.

### 4. Results

*4.1. Normality Test*

According to West, Finch, and Curran (1995), the criteria for the skewness coefficient is less than 2; the kurtosis coefficient should not be more than 7. In this study, the skewness was between −1.080 to 0.370, with a kurtosis between −0.720 to 1.919, suggesting that

the samples have fitted normality distribution. However, the composite reliability (C.R.) of multivariate kurtosis in this study was 60.93, failing to meet multivariate normality distribution. Therefore, the bootstrap proposed by Bollen and Stine (1992) was used to modify the overall model [73].

### 4.2. Common Method Variance (CMV Test)

This is a cross-sectional study, as all data were self-reported and collected within a certain period of time. The common method variance generated by measurement tools can affect systematic error and bias between theoretical constructs, and method variance can lead to overestimation or underestimation of correlations among the constructs, resulting in Type I or Type II errors. [75–78]. Detecting CMV involves putting all scales in the questionnaire into factor analysis of SPSS and choosing the principal component analysis with the maximum variation (VARIMAX) axis to determine CMV.

If common method variance exists, it is possible that: (1) a latent factor covers all of the variance; and (2) the first dimension explains most of the variance. The result of this study showed that the variance was 45.2%, which is lower than 50%, indicating that there was no common method variance in the constructs of this study.

### 4.3. Data Analysis for Validity and Reliability

This study used CFA to test the compatibility of each variable. The criterion for determining factor compatibility is based on the C.R. criterion proposed by Sparkman and Richard (1979) [79], which is above 0.7. According to Fornell and Larcker (1981) [80], the average variance extracted (AVE) value must be above 0.5.

The results in Table 1 indicate that each Cronbach's alpha ($\alpha$) $\geq$ 0.8, C.R. $\geq$ 0.7, and AVE $\geq$ 0.5, indicating that the measurement of this study has good validity and reliability.

**Table 1.** Confirmatory factor analysis.

| Variables | Items | Factor Loading | Cronbach's Alpha ($\alpha$) | C.R. | AVE |
|---|---|---|---|---|---|
| GE | I think participating in the Starbucks Rewards program is challenging. | 0.80 | 0.81 | 0.81 | 0.59 |
| | Participating in the Starbucks Rewards program gives me a sense of accomplishment. | 0.86 | | | |
| | I can receive membership information through the Starbucks app. | 0.62 | | | |
| CE | I spent a lot of money on participating in the Starbucks Rewards program. | 0.70 | 0.85 | 0.85 | 0.53 |
| | I have tracked the activities of Starbucks Rewards program. | 0.84 | | | |
| | The activities of Starbucks Rewards program (e.g., discounts, advertisements, etc.) get my attention. | 0.70 | | | |
| | If I have a problem operating Starbucks Rewards program, I contact with Starbucks' departments and seek assistance. | 0.67 | | | |
| | I will introduce Starbucks Rewards program to my friends. | 0.71 | | | |
| PP | When I participate in Starbucks Rewards program, I often forget the work I have to do. | 0.70 | 0.90 | 0.90 | 0.65 |
| | It makes me feel happy when I participate in Starbucks Rewards program. | 0.82 | | | |
| | It makes me feel interested when I participate in Starbucks Rewards program. | 0.65 | | | |
| | I would lose track of time when I am absorbed in Starbucks Rewards program. | 0.92 | | | |
| | It makes me curious when I'm using Starbucks Rewards program. | 0.91 | | | |
| ATT | I think the brand Starbucks can be trusted. | 0.56 | 0.85 | 0.85 | 0.60 |
| | I would choose Starbucks branded products over unbranded products. | 0.77 | | | |
| | Buying coffee-related products, the brand Starbucks is my first choice. | 0.86 | | | |
| | I consider myself a loyal consumer of the Starbucks brand. | 0.87 | | | |
| BI | I will keep buying Starbucks products. | 0.89 | 0.91 | 0.91 | 0.66 |
| | When someone asks for a recommendation, I would recommend the Starbucks app. | 0.87 | | | |
| | I want to give the Starbucks app a high online rating. | 0.73 | | | |
| | I will share the information on the Starbucks app on social media. | 0.83 | | | |
| | I will recommend the Starbucks app to my family and friends. | 0.74 | | | |

Note(s): CE: customer engagement; PP: perceived playfulness; ATT: attitude; BI: behavior intention.

### 4.4. Discriminant Validity Test

As shown in Tables 1 and 2, the AVE values corresponding to the MDE components, customer engagement, perceived playfulness, attitude, and behavior intention are 0.59, 0.53, 0.65, 0.60, and 0.66, respectively. According to Fornell and Larcker (1981), the square root of the AVE should be at least 0.5. The findings indicate that each variable had discriminant validity [80].

**Table 2.** Discriminant validity test.

| Variables | GE | CE | PP | Attitude | BI | AVE |
|---|---|---|---|---|---|---|
| GE | 0.77 | | | | | 0.59 |
| CE | 0.73 * | 0.73 | | | | 0.53 |
| PP | 0.67 * | 0.72 * | 0.81 | | | 0.65 |
| ATT | 0.58 * | 0.59 * | 0.54 * | 0.77 | | 0.60 |
| BI | 0.65 * | 0.70 * | 0.55 * | 0.76 * | 0.81 | 0.66 |

Note(s): MDE: mechanics, dynamics, and emotions; CE: customer engagement; PP: perceived playfulness; ATT: attitude; BI: behavior intention. * $p < 0.05$.

The second measurement for discriminant validity used in this study was the Heterotrait-monotrait (HTMT) ratio of correlation. The threshold of HTMT values is 0.85 [13]; if the value of HTMT is higher than 0.85, it can be considered to have a lack of discriminant. However, Gold et al. [81] proposed a value of 0.90. As shown in Table 3, the HTMT values of each variable are below 0.90, indicating that each variable has discriminant validity.

**Table 3.** HTMT values of each variable.

| Variables | GE | CE | PP | ATT | BI |
|---|---|---|---|---|---|
| GE | - | | | | |
| CE | 0.87 | - | | | |
| PP | 0.79 | 0.83 | - | | |
| ATT | 0.71 | 0.68 | 0.60 | - | |
| BI | 0.75 | 0.80 | 0.62 | 0.88 | - |

Note(s): CE: customer engagement; PP: perceived playfulness; ATT: attitude; BI: behavior intention.

### 4.5. Demographics of Respondents

In this study, most of the respondents are females, below 22 years old, and college students with monthly incomes below TWD 20,000, as shown in Table 4.

**Table 4.** Profile of respondents.

| Demographics | Respondents (%) | Demographics | Respondents (%) |
|---|---|---|---|
| Gender | | Occupation | |
| Male | 29.1 | Students | 64.5 |
| Female | 70.9 | Military personnel, civil servants, and teachers | 6.5 |
| Age | | Business | 7.6 |
| Below 22 | 63.0 | Freelance | 3.1 |
| 23–30 | 19.6 | Professional (Accountant, lawyer, doctor) | 2.8 |
| 31–40 | 5.0 | Service industry | 12.2 |
| 41 and above | 12.4 | Retired | 1.4 |
| Education | | Others | 1.9 |
| High School | 5.9 | Monthly income (TWD) | |
| College degree | 79.3 | Below 20,000 | 53.2 |
| Post Graduate degree | 14.8 | 20,001–30,000 | 19.3 |
| | | 30,001–40,000 | 9.8 |
| | | 40,001–50,000 | 5.3 |
| | | 50,001 and above | 11.9 |

*4.6. SEM Analysis*

The SEM fit measure included absolute fit indices, comparative fit indices, and Parsimony fit indices. This study referred to Kline (2011) for the model fit indices and recommended values for SEM analysis, and proposed the following fit indices for SEM, which are the Goodness-of-Fit Index (GFI), the adjusted Goodness-of-Fit Index(AGFI), the Root Mean Square Error of Approximation (RMSEA), the Normal Fit Index (NFI), Hoelter's CN (critical N), and the $\chi^{2/}$ df ratio (also known as the "normed chi-square") [82]. In the overall model fit analysis, GFI = 0.98, AGFI = 0.97, RMSEA = 0.01, NFI = 0.98, CN = 572.724, and $\chi^{2/}$ df = 1.02, indicating that the overall model has a good fit.

According to the results, the $R^2$ of customer engagement was 0.81, indicating that game elements (challenge, feedback, success) have 81% explanatory power on customer engagement. The $R^2$ of perceived playfulness was 0.72, indicating that game elements (achievement, challenge, feedback) have 72% explanatory power on perceived playfulness. On the other hand, the $R^2$ of attitude was 0.49, meaning that customer engagement and perceived playfulness have 49% explanatory power on attitude. The $R^2$ of behavior intention was 0.79, indicating that customer engagement and attitude have 79% explanatory power on behavior intention as shown on Figure 2.

The empirical results Table 5 support the hypotheses of this study. The relationship between game elements and customer engagement is positive and statistically significant ($r = 0.90$, $p < 0.05$), meaning that game elements enhance customer engagement. Thus, $H_1$ is supported. The relationship between game elements and perceived playfulness is positive and statistically significant ($r = 0.85$, $p < 0.05$), suggesting that game elements increase perceived playfulness. Thus, $H_2$ is supported. In addition, perceived playfulness and customer engagement are positively and statistically significant ($r = 0.19$, $p < 0.05$; $r = 0.55$, $p < 0.05$) with attitude, indicating that perceived playfulness and customer engagement can strengthen attitude. Thus, $H_3$ and $H_5$ are supported. The relationship between customer engagement and attitude is positively and statistically significant ($r = 0.42$, $p < 0.05$; $r = 0.58$, $p < 0.05$) with behavior intention, indicating that customer engagement and attitude promote behavior intention. Thus, $H_6$ and $H_7$ are supported. However, the empirical results do not support $H_4$. As shown in Figure 2, the relationship between perceived playfulness and behavior intention is not significant ($\beta = -0.04$, $p > 0.05$), which does not support our hypothesis (Table 5).

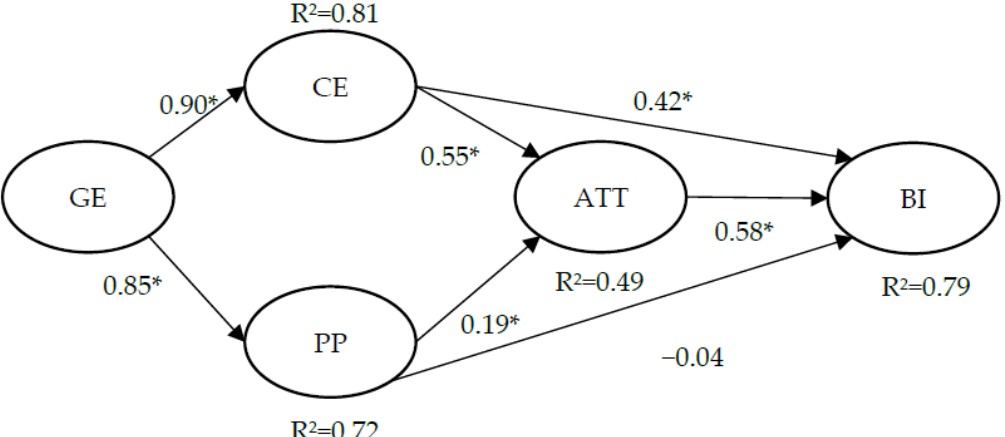

**Figure 2. SEM model with path coefficient.** Note(s): GE: game elements; CE: customer engagement; PP: perceived playfulness; ATT: attitude; BI: behavior intention. * $p < 0.05$.

**Table 5.** Results of hypothesis testing.

| Hypothesis | Path | Path Coefficient | Remarks |
|:---:|:---:|:---:|:---:|
| $H_1$ | GE→CE | 0.90 * | Support |
| $H_2$ | GE→PP | 0.85 * | Support |
| $H_3$ | PP→ATT | 0.19 * | Support |
| $H_4$ | PP→BI | −0.04 | Not Support |
| $H_5$ | CE→ATT | 0.55 * | Support |
| $H_6$ | CE→BI | 0.42 * | Support |
| $H_7$ | ATT→BI | 0.58 * | Support |

Note(s): GE: game elements; CE: customer engagement; PP: perceived playfulness; ATT: attitude; BI: behavior intention. * $p < 0.05$.

## 5. Discussion

Based on the MDE theory, the mechanism refers to game elements (challenge, feedback, success), emotion to perceived playfulness, and dynamics to customer engagement. This study constructed the proposed model based on the MDE theory to explore the influencing factors of Starbucks branded app on users' behavior intentions.

First, from the overall path analysis, this study found that game elements have a positive effect on customer engagement and perceived playfulness, which is consistent with Seaborn and Fels (2015), who maintained that gamification could motivate users to have a positive attitude, thus enhancing engagement [30]. Moreover, this study found that game elements play a significant role in human-computer interaction. It not only enhances the degree of engagement in human computers, but also brings pleasant feelings to users and positively affects subsequent behaviors.

Second, the results show that customer engagement is the critical factor in gamification. As indicated by the overall path analysis, customer engagement and perceived playfulness both positively influence attitude. This finding is consistent with Bergel, Frank, and Brock (2019), who maintained that customer engagement could create a positive attitude that would increase customer loyalty and positive perceptions [58]. Through using the Starbucks branded app, users' enjoyment and interaction can build their positive attitudes.

Finally, according to the overall path analysis, customer engagement and attitude have a positive effect on behavior intention, which is consistent with Alvarez-Milán et al. (2018), who suggested that engagement is essential to organizational behavior, marketing, social psychology, and education [56]. Customer engagement is a beneficial approach to prompt the individual's continued use of consumer/customer-to-user branded apps. More interaction between users and the app would effectively create a positive attitude and subsequent behaviors among users. Treiblmaier and Putz (2020) found that perceived playfulness positively influences attitudes and subsequent behavioral intentions [63]. However, this study found that perceived playfulness does not affect behavior intention, suggesting that perceived playfulness helps users to build a positive attitude, but does not directly affect their behavior. Moreover, though joyful experience in human-computer interaction helps users to build a positive attitude, but does not prompt users to use the app continuously.

## 6. Conclusions

Branded apps help companies to convey product information and strengthen their relationship with customers. By exploring the effects of game elements on perceived playfulness and customer engagement, this study found that game elements create positive emotion through perceived playfulness and customer engagement that prompt positive attitudes and subsequent behaviors. Furthermore, this study found that customer engagement is crucial to the success of brand marketing strategies.

### 6.1. Managerial Implications

This study found that game elements play an intrinsic motivation role, prompt human-computer interaction, and create a pleasant atmosphere. Game elements help users to

understand the functionality and entertainment of the Starbucks branded app, which could promote users' positive perception. The Starbucks branded app creates fun game elements (achievement, challenge, and feedback), to customers, which could strengthen the connection the Starbucks branded app with customers, and supported the membership of Starbucks rewards program, and the customer performance by using system.

Customer engagement has been recognized to drive customer behavior [57]. Vivek et al. (2012) pointed out that customer engagement is the connection between customers and products/services [70]. The Starbucks branded app is a platform for customers to show their loyalty. When the users go to the Starbucks stores, they would automatically use the app to collect stars. When the users go to the Starbucks stores, they would automatically use the app to collect stars. They would also receive notifications to remind them to complete the challenge. Such design of the app could strengthen the users' connection to the app, and motivate the company to develop marketing strategies to maintain customer relations. Through the app, Starbucks has created a strong bond with its customers, while enhancing customer emotion and behavior while they use the app.

Marino and Presti (2018) mentioned that the key success factor in the new relational marketing is customer engagement because it creates a lasting relationship with customers [83]. Therefore, this study suggests that the Starbucks branded app should add more activities, such as discounts and membership star upgrades, to strengthen the user relations.

According to Hwang and Choi (2020), in the gamification loyalty program, perceived playfulness positively affects attitude [52]. However, in this study, perceived playfulness has no significant effect on behavior intention. The branded app is another way to increase brand loyalty and sustainable consumption. Therefore, this study suggests that the Starbucks branded app should add more games to increase the fun and challenge of the app, and give more discounts.

### 6.2. Research Limitations and Future Research Suggestions

When an app is popular and mature, it should focus on user experience and utilize information technology into full play. However, the important factor for the usage of information technology, such as self-efficacy, perception of external control, and compatibility, were excluded from this study. Therefore, future studies could expand the research framework to understand the factors influencing the use of the Starbucks branded app.

Furthermore, this study only focused on users in Taiwan. In addition, the COVID-19 pandemic might have limited the distribution of questionnaires on social media. Although distributing questionnaires online could quickly spread to users in Taiwan, the demographics of the respondents were mostly limited to female college students. Therefore, future studies should expand the research scope and analyze the viewpoints of different groups.

**Author Contributions:** Conceptualization, C.-W.L.; methodology, C.-W.L.; software, C.-W.L.; validation, C.-W.L. and T.-Y.M.; formal analysis, C.-W.L. and C.-P.O.Y.; investigation, C.-W.L. and C.-P.O.Y.; writing—original draft preparation, C.-W.L. and C.-Y.C.; writing—review and editing, C.-W.L., C.-Y.C. and T.-Y.M.; visualization, C.-Y.C. and T.-Y.M.; supervision, C.-W.L. and T.-Y.M.; project administration, C.-P.O.Y. All authors have read and agreed to the published version of the manuscript.

**Funding:** This research received no external funding.

**Institutional Review Board Statement:** Not applicable.

**Informed Consent Statement:** Not applicable.

**Data Availability Statement:** Not applicable.

**Acknowledgments:** All subjects' enthusiastic participation is greatly appreciated.

**Conflicts of Interest:** The authors declare no conflict of interest.

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
