# Peer review of "Encouraging Sustainable Consumption through Gamification in a Branded App: A Study on Consumers’ Behavioral Perspective"

_sustainability, doi:10.3390/su15010589_

Round 1

Reviewer 1 Report

Dear authors, I first and foremost commend you on your relevant and innovative work on gamification as a drive for consumer behaviour, attitude, motivation, and engagement, which could prove of significant scholarly and practical use. 

The merits of the paper lie in the innovative character and adaptation to the spirit of the world we live in, its requirements and preferences, as well as with respect to the practical marketing results that a company may yield from the present research.

The information is backed by sources in the field, but their presentation is at times disparate and not always logically coherent, at times due to the quality of the English language. 

The statistics employed therein serve the quantitative research purpose of the paper and are presented in an orderly and clear manner followed by relevant results. 

I, therefore, consider that certain points need to be tackled extensively before the paper is ultimately published, should that be the case. My observations include:

1. There should be significant English editing, in the sense that the ambiguity and at times, the incorrect use of certain words, and word order shed an unfavorable light on the quality of the paper and leave room for interpretation and assumptions that should not find their way in an original scientific paper.

For instance, even starting from the abstract, which is the first perceived image of the paper for the reader, one can spot content mistakes that make the conveyance of information rather unclear- line 25- no subject (who? what?) and further one in lines 44 ( rise innovation program), 64-67, 197, 363, 413, 420-421, only to mention a few where the statements are not clear enough and they significantly impact on the quality of the paper. 

2. The proposition of the entire research endeavour is formulated in an unclear manner (see lines 113-114 and the use of the question mark). The hypotheses then follow in a broad manner but are suitably associated with work in the field and ultimately results and discussions. Perhaps the ultimate goal of the research associated with the hypotheses should be rephrased or mentioned in lines 113-114.

3. The research design and approach of research methodology on the respondents could be clarified further in terms of survey distribution, and sample shortcomings. 

4. The authors are kindly suggested to consider revisiting the Discussions parts, where for whichever reason (either incorrect use of English or interpretation of results), the clarity  is hindered (see line 363).

5. The authors could further consider the last paragraph of the Discussions part with the recommendations being expanded and instead included in the Conclusions part.

I wish you the best of luck with your work. 

Author Response

Dear Editors and Reviewers:

Thanks for the reviewers' comments concerning our manuscript entitled " Encourage sustainable consumption through gamification in branded app: A study on the behavioral perspective of consumer" (ID: sustainability-2002550). Those comments are valuable and helpful for revising and improving our paper and the critical guiding significance to our research. We have studied the comments carefully and have made a correction which we hope to meet with approval. Revised portions are marked in yellow on the revised manuscript. Please refer to the attached file for detailed authors' responses.

Sincerely,

Tso-Yen Mao Ph.D.

Department of Leisure Services Management, Chaoyang University of Technology

168, Jifeng E. Rd., Wufeng District, 413, Taichung, Taiwan, R.O.C.

Reviewer 2 Report

Thank you for giving me the opportunity to review the paper. I find the study interesting, but several claims and discussion are in need of revisions and support. Some of my constructive suggestions and comments are as follows:

1. Mention MDE and MDA completely before using the abbreviation in line 81 of page 2.

2. Line 144 page 3 is not a question.

3. From the statement "The two most significant factors in the model are perceived ease of 77 use and perceived usefulness, which are the core variables of TAM. However, Starbucks 78 branded apps include game elements, and original TAM can no longer meet this research 79 condition.", there is no explanation or basis why the 2 variables were removed. The claim should be supported and justified. I would argue that the claim to remove PEOU and PU could still be considered since these variables have been widely assessed when it comes to using technology and in evaluating its acceptance.

4. With the statement "The branded app is a marketing tool that retains engagement between brands and 105 customers. To provide an enjoyable customer experience in apps, companies should un- 106 derstand the factors that prompt or hinder consumers from using apps. In this study, we 107 believe that Starbucks branded apps have been using the game elements such as challenge, 108 achievement, feedback, emotion, etc., to be applied in a non-game situation to achieve 109 marketing concept.", I would assume that both PU and PEOU should be evaluated still since users are using the application and may hinder their continuous usage or acceptance. 

5. Your conceptual framework should be presented before the hypotheses building.

6. In accordance, the initial SEM should be presented for better visual representation of the results.

7. "Also, the use 284 of AMOS to adapt this SEM method is very appropriate as the analysis performed will 285 provide more accurate results." this claim should be supported by related studies.

8. Tests for normality before SEM should be conducted to provide readers the notion that the data can be used for SEM. In addition, the questionnaire should be provided in the manuscript.

9. Common Method Bias should also be conducted to provide justification of the latent constructs and items. 

10. Heterotrait-Monotrait Ratio should be performed to support the FLC findings for discriminant validity.

11. Where is the model fit to justify the SEM utilized in the study? 

12. The limitation of the paper should be expanded along with the suggestion for future research. With the current claim, there are a lot of lacking information that would help in promoting further research.

13. Practical and theoretical implications should be added.

14. Managerial insights should be highlighted to indicate how the findings can be applied in real-life. The current study has low novelty without the suggestions and application of the findings.

15. Conclusion should be expanded to cover all aspects done in the research.

16. I suggest the paper to run through a native English speaker before resubmission of revised manuscript.

Thank you and best regards to the authors. 

Author Response

(The authors gave the same response as above.)

Round 2

Reviewer 1 Report

Dear Authors,

I have carefully read your updated version and meets the requirements provided by the reviewers. 

Yours sincerely, 

Author Response

[2022.12.07]

Dear Reviewer:

This manuscript has been revised again in English by professionals. For detailed revision records, please refer to "Tracking revisions ."In addition, the reference part of this manuscript is edited with Endnote software. The document format is also amended according to the requirements of MDPI. Thanks again to the Reviewer for their enthusiastic assistance in reviewing the manuscript.

Sincerely,

Tso-Yen Mao. Ph.D.

Reviewer 2 Report

The authors were able to apply the suggestions. Thank you very much. 

Author Response

(The authors gave the same response as above.)
